# Who infects whom?—Reconstructing infection chains of *Mycobacterium avium* ssp. *paratuberculosis* in an endemically infected dairy herd by use of genomic data

**Annette Nigsch**[ID][1]*, **Suelee Robbe-Austerman**[ID][2], **Tod P. Stuber**[2], **Paulina D. Pavinski Bitar**[3], **Yrjö T. Gröhn**[3], **Ynte H. Schukken**[1,4]

**1** Department of Animal Sciences, Wageningen University, Wageningen, The Netherlands, **2** USDA APHIS National Veterinary Services Laboratories, Ames, Iowa, United States of America, **3** Department of Population Medicine and Diagnostic Sciences, College of Veterinary Medicine, Cornell University, Ithaca, NY, United States of America, **4** Royal GD, Deventer, The Netherlands

\* annette.nigsch@wur.nl

**Data Availability Statement:** All relevant data are within the manuscript and its Supporting

## Abstract

Recent evidence of circulation of multiple strains within herds and mixed infections of cows marks the beginning of a rethink of our knowledge on *Mycobacterium avium* ssp. *paratuberculosis* (MAP) epidemiology. Strain typing opens new ways to investigate MAP transmission. This work presents a method for reconstructing infection chains in a setting of endemic Johne's disease on a well-managed dairy farm. By linking genomic data with demographic field data, strain-specific differences in spreading patterns could be quantified for a densely sampled dairy herd. Mixed infections of dairy cows with MAP are common, and some strains spread more successfully. Infected cows remain susceptible for co-infections with other MAP genotypes. The model suggested that cows acquired infection from 1–4 other cows and spread infection to 0–17 individuals. Reconstructed infection chains supported the hypothesis that high shedding animals that started to shed at an early age and showed a progressive infection pattern represented a greater risk for spreading MAP. Transmission of more than one genotype between animals was recorded. In this farm with a good MAP control management program, adult-to-adult contact was proposed as the most important transmission route to explain the reconstructed networks. For each isolate, at least one more likely ancestor could be inferred. Our study results help to capture underlying transmission processes and to understand the challenges of tracing MAP spread within a herd. Only the combination of precise longitudinal field data and bacterial strain type information made it possible to trace infection in such detail.

## Introduction

*Mycobacterium avium* ssp. *paratuberculosis* (MAP) is the causative agent of Johne's disease, or paratuberculosis, a chronic, slowly progressing disease of ruminants associated with high

Information files. In addition, Sequence read data have been deposited at NCBI in the Sequence Read Archive database under the following two BioProject accession numbers: PRJNA725521 and PRJNA686527.

**Funding:** This work was supported by the USDA-NIFA AFRI [grant number # 2014-67015-2240] as part of the joint USDA-NSF-NIH-BBSRC-BSF Ecology and Evolution of Infectious Diseases program (main applicant: YG). The funders had no role in study design, data collection and analysis, decision to publish, or preparation of the manuscript.

**Competing interests:** The authors have declared that no competing interests exist.

economic losses, especially in dairy herds. Challenges in the surveillance and control of MAP are a long incubation period of 1–15 years [1], and inefficient diagnostic tests, which lead to limited success of control programmes. The role of MAP in the pathogenesis of Crohn's disease in humans is still controversial [2].

The primary route of MAP infection is faecal-oral by direct or indirect contact with the pathogen. Calves are highly susceptible during the first weeks after birth, and resistance to infection increases until one year of age [3]. Calves become infected either horizontally or vertically (*in utero*). Transmission from dams to calves at an early age is currently regarded as the most important route of infection and is therefore the focus of many control programmes. In adults, ingestion of MAP does not necessarily lead to infection, but repeated uptake of high doses of bacilli may result in adult infection [4, 5]. Adult-to-adult, calf-to-calf and heifer-to-heifer infections have been shown to exist [4, 6–9]. These routes typically receive little attention in MAP control programmes.

Large differences in MAP shedding patterns can be observed. Intermittent shedders, low shedders ($\leq$50 colony-forming units per gram (cfu/g) faecal matter), high shedders ($>$50–$10^4$ cfu/g faecal matter), and super-shedders ($>10^4$ cfu/g of faecal matter) are known shedding categories for individual animals. The majority of cows will never develop high shedding levels, since many cows never reach advanced enough age [10]. Schukken et al. found two distinct infection patterns, so called progressors and non-progressors [8]. Progressors are characterised by continuous and progressive shedding of high MAP loads and high antibody production. Non-progressors present intermittent and low shedding of MAP bacteria and a virtual absence of a humoral immune response, suggesting that they have the infection process under control. Building on these findings, Mitchell et al. [10] distinguished between two categories of progressors, linked to immune control and the age at onset of shedding: cows that start shedding at a younger age partially control the infection, but eventually become high shedders (slow progressive infection), while cows that start shedding persistently at an older age progress rapidly with shedding and lack effective control of infection. Obviously, super-shedders represent the greatest risk for spreading MAP among herd mates [11]. However, removing high-shedding animals (which are easily detected) has shown to be insufficient to address long-term persistence of MAP [12, 13]. Simulation models have given further support to the hypothesis that intermittent, low and transiently shedding animals play an important role in maintaining low prevalent infections in dairy herds [14]. Quantitative estimates of the importance of transmission routes at all ages of the host and of the role of animals presenting these different shedding patterns are essential to decide on relevant control procedures.

The MAP genome is extremely stable with an estimated mutation rate $\mu$ of the core genome of one mutation per 2–7 years [15]. Earlier literature assumed clonal infections of herds with a single strain of MAP bacteria [16, 17]. However, several studies now have shown that multiple strains of MAP may be simultaneously present in a herd [7, 18], suggesting that several concurrent infection cycles within a single population are possible. More recent studies even demonstrated the incidence of a mixed, simultaneous infection by three, or even up to five genotypically diverse MAP isolates in a single dairy cow [19, 20]. Within the individual host, the MAP population is initially thought to be genomically homogeneous, but will diversify over time due to mutations. These processes of within-host evolution of MAP and mixed genotype infection of hosts with multiple MAP strains need to be considered in further studies to draw valid conclusions about the complexities of MAP transmission [21, 22]. For such studies sequencing a single isolate from each case was suggested to be inadequate in the presence of within-host diversity, but frequent sampling will improve accuracy [23]. For MAP it is currently not known whether the low mutation rate will allow detailed analyses of infection chains. With whole genome sequencing (WGS) data the highest possible degree of

discrimination between pairs of isolates can be achieved. Nevertheless, integration of non-WGS data into analysis of transmission pathways is suggested to lead to considerable refinement in our understanding of the epidemiology of mycobacterial disease [21].

With falling costs of large-scale genome sequencing and advances of biostatistical tools, population genomic studies are increasingly used to study pathogen spread within populations. Traditionally, network inference models were used to identify transmission chains in early stages of disease outbreaks. In endemic settings, network inference faces multiple challenges, such as: (a) non-sampled early generations of cases and thus uncertainty about which of the sampled strains is genomically closest to the originally introduced strain and can thus be considered as the most recent common ancestor; (b) multiple introductions of genomically diverse strains over time, resulting in a polyphyletic sample; and (c) as a consequence of (b), exposure of hosts to multiple strains which may lead to mixed genotype infections.

This study aims to identify individual animal-to-animal infection chains ("who infects whom"), in order to better understand the infection dynamics of MAP in endemically infected dairy herds. Transmission trees will be constructed by using WGS data in combination with detailed longitudinal epidemiological data. Support of the reconstructed infection chains for the current prevailing hypotheses on transmission routes will be evaluated, and the role of individual animals in infection spread will be investigated. In addition, within-host and within-herd diversity of MAP will be characterised to provide fundamental input to all advanced analyses. To conclude, it shall be discussed whether observational field data are precise enough to perform relevant analyses to inform future research.

## Methods

### Study population

**Data collection.** Longitudinal data from an endemically infected MAP dairy herd in New York State in the northeast United States were collected over eight years. The dairy herd consisted of approximately 330 cows. Johne's disease status of individual cows was determined *ante mortem* through biannual faecal and quarterly serum sampling. Sampling of cows started at first calving. An additional 170 cull cows could be tracked to the abattoir, where four gastro-intestinal tissues and a faecal sample were collected from each cow *post mortem*. The harvested tissues included two lymph nodes located at the ileocecal junction and two pieces of ileum, one taken from 20 cm proximal to the ileocecal valve and the other taken from very near the ileocecal valve. *Ante mortem* sampling commenced in February 2004 and continued until October 2010, and the last abattoir samples were taken in 2011. In total, 2.7% (114/4,158) faecal samples, 24.0% (149/621) tissues and 1.5% (89/5,937) serological samples from 1,056 individual cows were MAP positive. The farm environment was sampled in approximately 20 locations on a biannual basis, resulting in 14.8% (34/230) positive bacterial cultures. In addition, precise demographic data–including birth date, birth pen location, calving dates, fertility data, animal pen locations, dry-off dates and eventually culling information and cull dates–were collected during 1988–2012. For a more complete description of sampling, see Pradhan et al. [7] and Schukken et al. [8]. Ethical approval was not required, as all samplings took place as part of an ongoing MAP herd control programme.

The selection of the study population for this research was done retrospectively and was based on the availability of sequenced MAP isolates. Accordingly, the study population consisted of all MAP positive cows (n = 66) and all MAP positive environmental samples (n = 22) from which MAP isolates (n = 150) could be successfully sequenced.

**Farm management and MAP control.** The farm participated in a MAP control programme, was well managed and had a good hygiene status. It was a closed farm for years

before the start of the study and did not purchase animals during the study. Apparent herd-prevalence based on bacterial culture (faecal matter and tissues) and serology was as high as 7.6% in 2004 and decreased to 0–2.4% in 2010 [24]. Throughout the study, the farm owner was informed about all test results and advised on optimal management practices to reduce MAP prevalence. In terms of management groups, youngstock and cows were transferred among 14 different locations: individual calf hutches, six calf and heifer rearing pens, and seven freestall pens for cow groups in high and low lactation, including separated maternity pens and sick pens. In 10–30% of calvings, maternity pens were used for more than one cow. Calves were separated immediately after birth and were not allowed to nurse cows. Colostrum fed to calves was from MAP-negative tested cows. Youngstock were not kept near adults, but indirect contact was possible via employees. Bred heifers were housed on another nearby facility. Animal pen location data were kept accurately so that animal location was reliably available on a daily basis. Based on these pen location data a social network with number of days with direct pen contact between pairs of cows was established for the subset of 66 cows. Pairs in this subset of MAP-shedding cows had on average 142 contact days during their lives (median: 84; max: 1167). For 27% of pairs of cows no direct pen contact was recorded. These were in particular cows born in different years and cows that calved around 6 months apart from each other. The dataset contained sequences from four dam-daughter pairs. Nine cows were super-shedders and seven cows were progressors; these cows were culled between 6–29 months after their first positive MAP test. The decision when to cull was taken by the farm owner based on economic reasons.

## Laboratory analysis, strain sequencing and genotyping

Faecal samples, tissues and environmental samples were cultured in Herrold's egg yolk media (HEYM) for up to 16 weeks at 37˚C and shedding levels (cfu of MAP/tube) were determined. Each culture with colony growth was sub-cultured. DNA was extracted from single bacterial colonies sub-streaked on HEYM slants. The analytical protocol for bacterial culture was described in detail by Pradhan et al. [7]. For DNA extraction, strains were grown for 12 weeks at 37˚C in Middlebrow 7H9 broth with 10% Middlebrook OADC, 0.05% tween 20, 1ug/ml micobactin J, and 0.01% cyclohexamide from multiple colonies picked of HEYM slants. Epicentre's MasterPure Gram Positive DNA extraction kit was used with the addition of a 20min 80C incubation prior to lysis. DNA was prepared for sequencing with the Nextera XT DNA library kit and sequenced using Illumina HiSEQ 2500 2x100 paired end rapid run. The sequencing was conducted as part of the current study. Analysis of WGS sequences was performed using vSNP, National Veterinary Services Laboratory's in-house single nucleotide polymorphisms (SNP) detection pipeline [25]. Briefly, the Illumina sequence reads for each isolate were mapped to the reference genome MAP K-10 using the Burrows Wheeler Aligner [26] and Genome Analysis Toolkit (GATK) [27–29]; according to GATK best practices. Integrated Genomics Viewer was used to visually validate SNPs. The final SNP alignment contained 150 sequences of 1,472 SNPs of the core genome, with collection dates ranging from 17th February 2004 to 5th March 2008 (data published in S1 File). For detailed genomic statistics see Richards et. al [30]. Isolates were bio-banked at the Department of Clinical Studies, University of Pennsylvania, School of Veterinary Medicine, New Bolton Center, Kennett Square, PA 19348, USA, and are available on request. Sequence read data have been deposited at NCBI in the Sequence Read Archive database under the following two BioProject accession numbers: PRJNA725521 and PRJNA686527. For unique identifiers to link the SNP alignments with the sequence read data, see S1 Table.

## Analysis of strain diversity

Strain diversity was estimated at cow-level (within-host strain diversity) and at herd-level (within-herd strain diversity). Isolates sampled from the same cow were compared to isolates shed by other cows. As a measure of genomic diversity, pairwise distance was calculated based on the number of SNPs between each pair of isolates. Isolates with zero SNP differences are referred to as isolates with identical genotype; isolates that differ by at least 1 SNP are referred to as different genotypes. MEGA7 was used to estimate the maximum likelihood phylogeny [31].

## Reconstruction of transmission trees

To reconstruct within-herd transmission trees, a phylogenetic network analysis was performed with an algorithm called SeqTrack [32], implemented in the *adegenet* package [33] in R [34]. This algorithm is a graph-based approach recovering maximum parsimony phylogeny to identify the most likely ancestries from aligned core SNPs in pathogen genomes. Jombart et al. [32] based their method on three observations: 1) each sampled isolate will only have one unique ancestor (in the absence of recombination and reverse mutations), 2) descendants will always follow their ancestors in time, and 3) among all possible ancestries of a particular isolate, some are more likely than others, and this likelihood of ancestry can be estimated from the genomic distance between sampled isolates. In situations where several potential ancestors may exist for a given isolate, additional rules are needed: the best ancestor will be selected by adding proximity information in the form of weighting matrices. This rule is particularly relevant for slowly evolving pathogens where even a long-term, endemic setting may result in low genomic diversity. These are indeed characteristics of endemic MAP infections in dairy herds. In addition, SeqTrack's flexibility to incorporate various types of epidemiological information with a number of different weighting matrices was judged to be beneficial for the analysis of our detailed longitudinal cohort data as it allowed investigation of the additional value of epidemiological data in the reconstruction of transmission trees.

Inference of ancestry follows a strict hierarchy: 1) temporal order: isolates with the earliest dates are at the root, and those with the latest dates are at the tips of the reconstructed tree: we used two different dates: inferred start of shedding and birth date of cow; see "Scenarios", 2) genomic distance: based on the number of SNP differences between pairs of isolates (entered as distance matrix), 3) epidemiological weight: see "scenarios", and 4) probability $p$ of observing a given number of mutations between an isolate and its ancestor: $p$ was computed based on the mutation rate $\mu$ of the pathogen (0.25 substitutions/core genome/year), time interval between each pair of isolates, and length of partial nucleotide sequences (1,472 SNPs), using maximum likelihood. As the genomic distance between two isolates $a$ and $b$ increases, $p$ decreases that $a$ is the direct ancestor of $b$. SeqTrack only relies on epidemiological weights and $p$ to resolve ties in the choice of ancestry: if an isolate has more than one potential ancestor in identical genomic distance, the ancestor with the higher weight is assigned. If two potential ancestors have the same weight, the ancestor with the higher $p$ value is assigned. The analysis is thus largely insensitive to $\mu$.

## Extension of SeqTrack to endemic infection

A number of extensions were made in this work to take SeqTrack a step forward to derive individual infection chains for endemic infection with characteristics of MAP.

**Epidemiological unit.** SeqTrack was designed for epidemics where one single isolate is sampled from each case and cannot capture within-host pathogen genomic diversity. To distinguish between mixed genotype infections and within-host evolution, the transmission tree

was built at isolate level (= epidemiological unit), instead of case (cow) level. Most supporting epidemiological data were collected at cow level. However, isolate specific parameters were sampling date, pen contacts of the cow at sampling day, and duration of exposure to other MAP-shedding cows.

Transmission events at isolate level are referred to as ancestries between an ancestor and its descendant; at cow level, the terms source of infection and recipient are used. A source or recipient could be either another cow or an environmental sample.

**Duplicate genotypes.**   If the identical genotype could be isolated multiple times from the same cow, duplicate sequences (n = 22) were discarded. Continuous shedding from the earliest to the last sampled isolate of this genotype was assumed and the infectious period was set as described in the next section. A total of 128 isolates were included in the reconstruction of transmission trees.

**Scenarios.**   Temporal order of sampling can be misleading owing to variable delays between exposure and sampling, even for MAP with such a slow rate of mutation. The challenge of determining the probable time window when MAP-positive cows became infected and infectious was addressed by comparing six infection scenarios. We hypothesize that if several scenarios resulted in the same choice of ancestor for a given isolate, this would add support to the accuracy of reconstructed transmission chains.

**Scenario [Basic]—Basic transmission tree:** based only on genomic distance (without any weighting).

**Scenario [E]—Weighting by exposure time:** [Basic] plus weighting matrix with number of days cow *X* spent in the same pen with any other cow *Y* during cow *Y*'s infectious period before cow *X* started to shed (Fig 1). The number of days of exposure time [E] was calculated for each pair of isolates and entered in a matrix with 128 rows x 128 columns (one column and row for each isolate). The value of [E] was then used as the weight, with highest weights for longest exposure. For cows with several MAP genotypes, the duration of the infectious period and [E] were calculated for each genotype separately. The genotype-specific infectious period was defined as starting at the mid-day between last negative and first positive sampling date and ending at the mid-day between last positive and consecutive negative sampling date. The infectious period for abattoir samples ended the day before culling. The inferred mean infectious period was 95 days (min: 1 day for cows that tested negative the day before slaughter, median: 71, max: 779 days for cows shedding the same genotype serially at several sampling dates). Accordingly, the mean duration of [E] for pairs of isolates sampled directly from cows was 8 days (min–median–max: 0–0–419 days), with 82% of pairs of isolates with 0 days of [E].

Environmental samples were assumed to represent (potentially non-detected) infectious cows. Their "infectious period" of spill-back was therefore defined in the same manner as for cows, and its start was used as the date for both [birth] and [shed] scenarios (explained below).

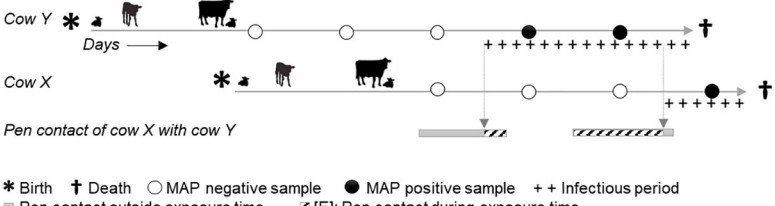

**✱** Birth   **†** Death   ○ MAP negative sample   ● MAP positive sample   + + Infectious period
▬ Pen contact outside exposure time   ▨ [E]: Pen contact during exposure time

**Fig 1. Example of contacts between two cows (*X* and *Y*) over time.** The exposure time [E] is the time cows *X* and *Y* have pen contact and cow *Y* sheds a certain MAP isolate, whereas cow *X* does not yet shed. The vertical arrows indicate the shedding starts of both cows. The shedding start of cow *X* corresponds to the end of [E]. Overall pen contact days: contact during and outside [E].

The environment served as a potential source of infection for all cows that were, during, the spill-back period, at the location where the environmental isolate was sampled. Average duration of spill-back of the 22 environmental samples was calculated to be 91 days (min–median–max: 42–91–116), and their mean duration of [E] was 26 days (min–median–max: 0–0–116 days), with 69% of pairs of environmental isolates and isolates sampled directly from a cow with 0 days of [E].

**Scenario [S]—Weighting by susceptibility:** [Basic] plus [E] plus additional weighting matrix to reflect the decreasing susceptibility of cows over time. Seven social network patterns (weights from 6 to 0) were used to weight potential transmissions based on age of the susceptible animal at contact and duration of its exposure. The weighting order was based on accepted knowledge of MAP epidemiology knowledge. Each pair of isolates was assigned one weight that reflected their epidemiological link:

6. **Cow-to-calf contact:** (direct or indirect) cow-to-calf contact within the first days of life of a newborn calf (maximum weight for isolates from own dam or any other cow present in the maternity pen that calved ± 15 days around birth date of the cow),

5. **Calf-to-calf contact:** direct contact in the first year of life of a cow within the same age cohort (other cows born ± 30 days around birth date of the cow),

4. **Adult-to-adult contact during the infectious period:** pen contact ([E] ≥1) during adulthood,

3. **Longer direct contact:** long pen contact during adulthood with cows outside their infectious period (≥100 pen contact days, but [E] ≥0),

2. **Limited direct contact:** limited pen contact during adulthood with cows outside their infectious period (1–99 pen contact days, but [E] ≥0),

1. **Indirect contact:** pairs of cows which lived on the farm during the same period, but with no recorded pen contact days, and

0. **No contact:** one cow was culled/sold before the other cow was born.

[Basic], [E] and [S] scenarios were each calculated with two different dates to account for the uncertainty of the temporal sequence of exposure times: [birth]: birth date of cow, and [shed]: potential start of MAP shedding and thus of the (genotype-specific) infectious period (for calculation of [shed] see [E] scenarios). These two dates were selected to investigate how ancestries change if susceptibility is put as the focus of infection dynamics (with scenarios using [birth]) versus infectiousness (with scenarios using [shed]). As infection spread is driven by both infection states, it was expected that [birth] and [shed] scenarios with [E] and [S] weights would result in more similar trees than [Basic] scenarios. In total, six scenarios were then calculated, namely [birth_Basic], [birth_E], [birth_S], [shed_Basic], [shed_E] and [shed_S].

SeqTrack will define the best fitting transmission tree based on maximum parsimony. The basic model without weights as described in these scenarios will logically provide the maximum parsimony model. The basic model is only based on genomic distances and does not take epidemiological limitations into account. A genomic connection between two isolates from two cows that in real life were never on the farm at the same time is acceptable in the basic model, but will receive a very low weight in the model expanded with epidemiological information. Therefore, maximum parsimony should only be compared within the same scenario or between scenarios when no conflicting epidemiological information is present.

**Within-herd circulation of genomically diverse strains.** SeqTrack assumes monophyletic genomic data (infection caused by a single external source) and will add all isolates into

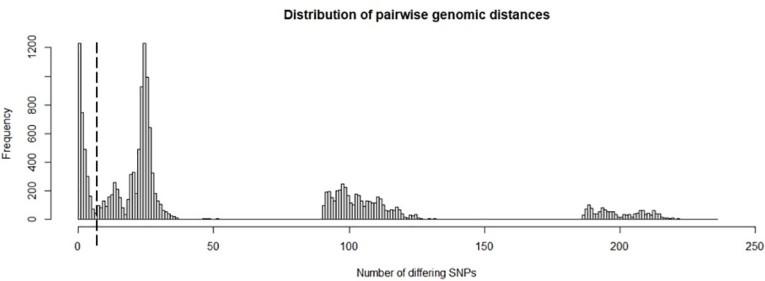

**Fig 2. Distribution of pairwise genomic distances (n = 150 isolates with 1,472 SNPs).** The dashed line marks the genomic distance threshold of 6 SNPs.

one single transmission tree, independent of how distant (and thus less likely) reconstructed ancestries may be. With the low mutation rate of MAP, strain diversification within the study period through evolution was expected to be limited. A genomic distance threshold of 6 SNPs was defined based on the overall herd-level genomic diversity (Fig 2). Pairs of isolates exceeding this threshold were considered not to have arisen from directly linked cases. If no ancestor within this threshold could be found in the sample for a particular isolate, it was set as the root of a separate transmission tree, indicating that the true ancestor had not been sampled. Generally, the lower a threshold of number of SNPs is chosen, the more transmission trees with multiple generations will be broken up into smaller individual trees or unconnected singleton isolates (resulting trees of a sensitivity analysis with different threshold values are not shown).

**Censored data.**   Non-sampled early generations of cases (before study start) would lead to overestimation of the number of descendants for isolates at the root of the transmission tree. In the absence of their true ancestor in the sample, SeqTrack assigns more descendants to the earliest sampled isolates. In addition, data were right censored, and the number of descendants were underestimated, particularly for isolates sampled in the late phase of the study. Whereas the transmission trees were reconstructed with all 128 isolates, the following conservative assumptions were made for the estimation of genotype-specific and cow-specific reproduction ratios to account for temporality in the data structure: For the earliest 10% of isolates only one third of the assigned descendants were assumed to be their true descendants. For the latest 10% of isolates the number of descendants per isolate were not calculated as these descendants were not yet fully sampled within the study period. Consequently, the role of individual cows in infection spread was analysed for the remaining 84 isolates, sampled from 57 cows.

**Lack of genomic resolution.**   If isolates with identical genotype can be sampled over years from generations of cows, a range of alternative infection chains may exist (Fig 3). In addition to the number of recipients according to the one, optimal tree of SeqTrack, the "potential number of recipients" was calculated for each cow, assuming that all isolates with identical genotype that fulfilled certain criteria could be descendants of the same cow. The criteria were those for [S] weights 4–6, as described under "Scenarios".

## Network analysis

**Analysis at isolate level.**   A MAP genotype-specific effective reproduction ratio $R_{GT}$ was calculated by aggregating the number of descendants per genotype at herd level. Reconstructed transmission trees resulting from all six scenarios were compared and changes in the branching of the trees due to epidemiological weightings were assessed. Ancestries of pairs of isolates were identified that were identical in two or more scenarios. By comparing the maximum likelihood $p$ of all individual ancestries across scenarios, overall statistical support for each scenario was assessed.

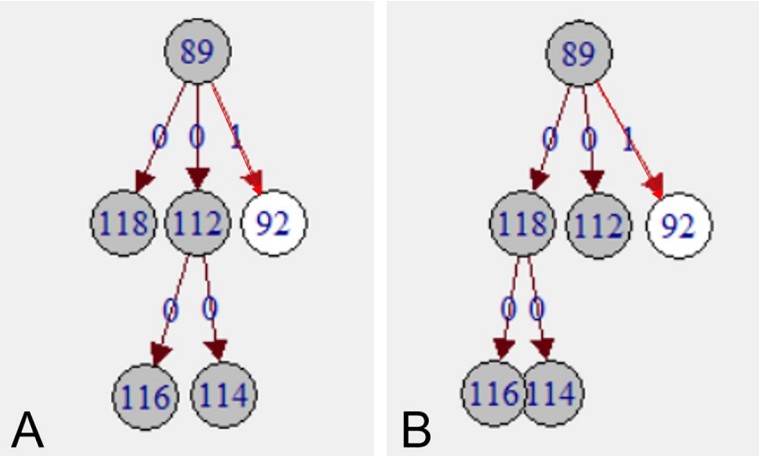

**Fig 3. Reconstructed transmission trees.** Transmissions are depicted by edges and isolates by vertices in a directed network. Edge labels and edge colour indicate number of SNPs of differences between ancestor and descendant. (A) and (B) two alternatives of potential infection chains of five isolates with identical genotype (dark grey). More alternatives exist.

**Analysis at cow level.** Number of recipients produced during lifetime (animal-specific effective reproduction ratio $R_A$), "potential" recipients and sources of infection per cow were quantified by summing up ancestors and descendants of isolates. Support of the reconstructed infection chains for the current prevailing hypotheses on transmission routes was evaluated by quantifying epidemiological links of ancestries across scenarios: all reconstructed ancestries were retrospectively matched with the seven social network patterns defined under [S].

Results on the number of recipients produced were validated against the literature on risk factors for MAP spread. For this purpose, associations between four *ante mortem* detectable Johne's disease phenotypes of the cows in the study population and their number of recipients were tested for all six scenarios. The four phenotypes were: **shedding level** (three levels: [0] faecal culture negative, [1] low (1–50 cfu/tube of faecal matter), [2] high (>50 cfu/tube of faecal matter)), **age at first shedding** (three levels based on Mitchell et al. [10]: [0] first positive faecal sample collected before the age of 3 years, [1] first positive faecal sample >3 years, [2] cow was *ante mortem* never positive), **infection progress** (Inclusion criteria: individual cows with at least four MAP culture results, and at least one faecal sample taken after a positive MAP culture. Three levels: [0] faecal culture negative, [1] non-progressor, [2] progressor), and **serostatus** (two levels: [0] no ELISA-positive test, [1] at least one ELISA-positive test). For shedding level, infection progress and age at first shedding correlation was calculated with Spearman's rank correlation. For the binary variable serostatus, the difference between mean number of recipients was calculated with Welch's two sample *t* test.

The R code for this analysis is published in the S2 File.

## Results

### Strain diversity within the host over lifetime (at animal-level)

Up to 8 MAP isolates could be sequenced per cow. Cows had up to 5 non-clonal MAP genotypes. For 43 cows with only one MAP isolate sampled, strain diversity could not be assessed. Out of 23 cows with 2–8 isolates, only two (8%) shed the identical genotype in series at different sampling days. Only from three (13%) cows could an identical genotype be isolated both *ante mortem* from faecal matter and *post mortem* from tissue; all three were super-shedders

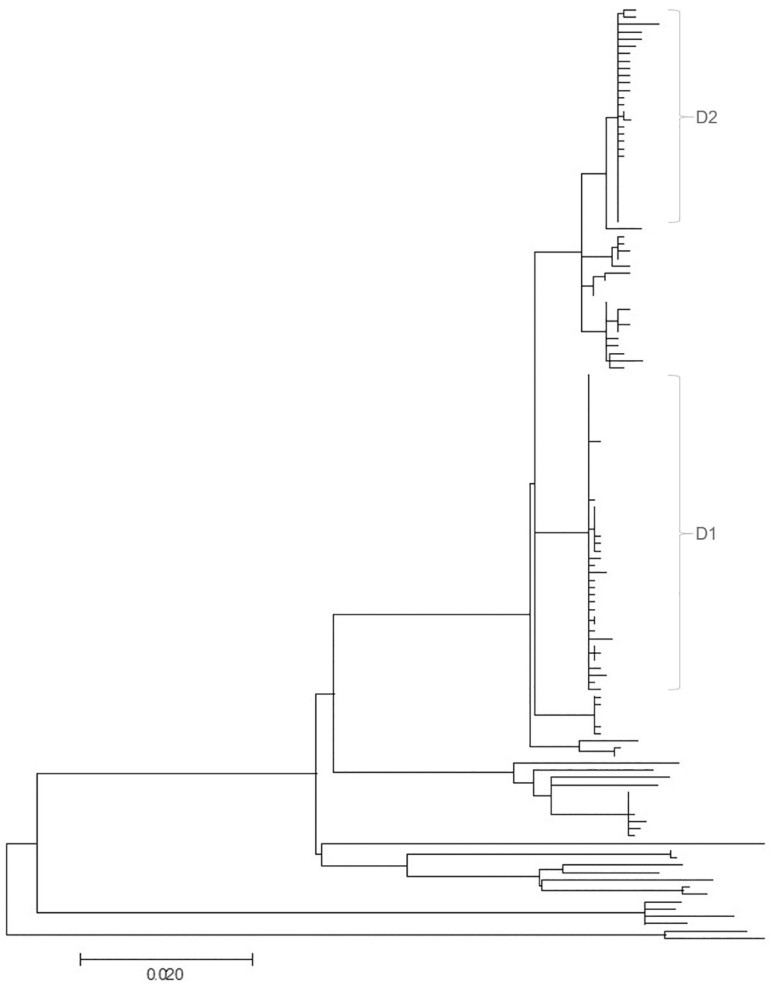

**Fig 4. Maximum likelihood phylogeny of sequenced MAP isolates.** Phylogenetic tree based on SNP data from 128 sequenced MAP isolates with 1472 nucleotide positions, using the Tamura-Nei model. A total of 94 different genotypes were recovered. The tree with the highest log likelihood (7449.75) is shown. Branch lengths is measured in the number of substitutions per site. D1 and D2 indicate isolates belonging to the two dominant strains.

and progressors. MAP positive tissue–confirming true infection–contained 1–3 different genotypes per cow. For eight (35%) cows, all isolates differed by a maximum of 6 SNPs. The remaining 15 cows had at least one isolate with 7–234 SNPs of difference, indicating mixed infection.

## Strain diversity within-herd (at herd-level)

A total of 94 different genotypes were recovered from all 150 sequenced isolates (Fig 4); of these 84 (89%) were only detected once. The most prevalent genotype was isolated 32 times from twelve cows and four environmental samples. Several MAP strains with genomic distances of more than 100 and 200 SNPs between strains were recorded (Fig 2). With an estimated mutation rate of one substitution per 2–7 years, a genomic distance of 100 SNPs indicates multiple introductions of MAP strains rather than within-herd evolution from a common ancestor. Two dominant strains (D1 and D2) of MAP could be detected during the whole study period. These dominant strains were responsible for 19% (D1) and 35% (D2) of MAP infections. Dominant strains consisted of clusters of 24 (D1) and 21 (D2) genotypes that

differed by maximally 4 and 7 SNPs, respectively. Each dominant strain had one "super-spreading" genotype which resulted in 27 (D1) and 31–33 (D2) descendants, depending on the scenario. Averaged over the whole cluster of genotypes, the $R_{GT}$ of the dominant strains were 1.2 (D1) and 1.7–1.8 (D2). Genotypes that did not belong to the dominant strains had an average $R_{GT}$ of 0.4.

## Reconstructed transmission trees

Transmission trees showed similar features in all six scenarios: two main trees, a range of small trees with 2–4 generations and singular, unconnected isolates. These unconnected isolates were not (closely) related to any other isolate sampled on this farm and there was no indication that they spread during the study period within the herd. Main trees were formed by a cluster of genotypes of one of the dominant strains. Remarkable features of main trees were long branches of infection chains with isolates of identical genotype, and 2–4 super-spreaders (each with 10–20 descendants), plus variable numbers of isolates with 2–6 descendants. Characteristics of these features differed considerably between scenarios (Fig 5; for [shed] scenario trees

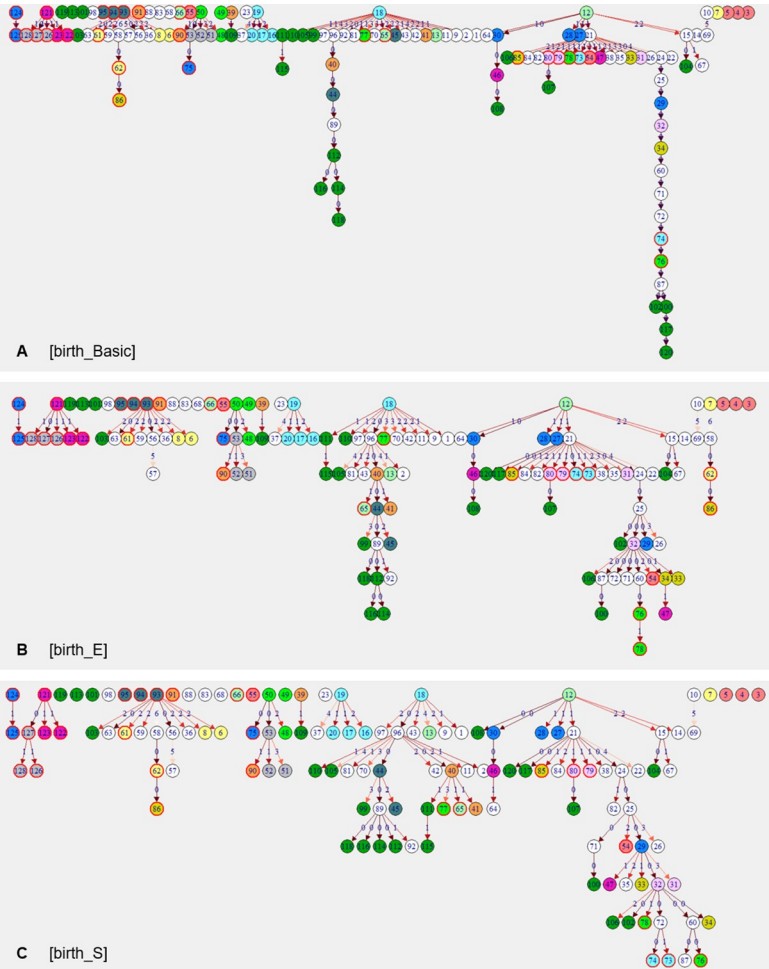

**Fig 5. Reconstructed transmission tree of three scenarios (n = 128 isolates).** (A) [birth_Basic], (B) [birth_E], (C) [birth_S]. Isolates sampled from the same cow are labelled with successive numbers and are shown in vertices of the same colour and outline. White vertices represent cows with only one isolate. Dark green vertices (labelled 99–120) represent environmental samples. Edge labels and edge colour indicate number of SNPs of difference between ancestor and descendant. For a list of descendants, their ancestors and the pairwise genomic distance, see S2 Table.

see S1 Fig): two long branches with 8 and 15 generations of isolates with identical genotype and super-spreaders at or close to the root were particularly prominent in [Basic] scenarios. These long infection chains had short time-intervals between generations of descendants of a few weeks to months, untypical for MAP. [E] and [S] scenarios resulted in more branched trees, less prominent super-spreaders, and more isolates serving as ancestors for up to 6 descending isolates. Infection chains in [E] and [S] scenarios had a maximum of 7–8 generations, indicating longer time-intervals between two transmission events. Individual transmission trees were composed of exactly the same isolates in [birth] and [shed] scenarios, but in different orders.

Thirty-eight (30%) isolates were assigned the identical ancestor in all six scenarios, and for 6 (5%), 18 (14%) and 49 (38%) isolates the algorithm returned five, four or three times with an identical ancestor, respectively. No isolate was assigned a different ancestor in every scenario, and for each isolate, at least one ancestral isolate could be identified with more statistical and/ or epidemiological support. The three [birth] scenarios and the three [shed] scenarios showed more identical ancestries amongst themselves (64–80%) than [birth] compared to [shed] scenarios (45–61% identical ancestries). [birth_E] and [shed_E] shared 61%, [birth_S] and [shed_S] shared 60%, and [birth_Basic] and [shed_Basic] shared 48% of ancestries. Scenarios with weighting for exposure time and susceptibility were thus closer to each other than [Basic] scenarios without any epidemiology incorporated. Overall statistical support was numerically highest for [birth_Basic], followed by [birth_E], [birth_S], [shed_Basic], [shed_S] and [shed_E]. [Basic] scenarios will, by definition, have the highest $p$ values, as they represent the most optimal genomic tree, but they lack epidemiological support. Epidemiological weighting incorporated in [E] and [S] can only deviate from this optimal tree, but outweighs loss of numerical credibility by epidemiological reliability. Differences in statistical support were small (all scenarios had 34–35 ancestries with $p > 0.95$; and the average $p$ of ancestries ranged from 0.34–0.35 across all six scenarios), indicating that epidemiology-informed scenarios had similar statistical support as [Basic] scenarios (data published in S2 Table).

Adult-to-adult contact during the infectious period was in all scenarios except [birth_Basic] the most frequent social network pattern leading to infection, followed by direct contact during adulthood outside the infectious period and indirect contact (Table 1). Remarkably, cow-to-calf contact was the least important transmission route (0–4% or 0–4 ancestries in each scenario) and was even less frequent than ancestries with no epidemiological link at all. Genomic

**Table 1. Ranking of transmission routes.** Ranking of transmission routes by the proportion of inferred ancestries based on social network patterns. Column percentages add up to 100%. The overall rank was inferred from the sum of ranks of all scenarios.

| Weight | Description | Scenarios | | | | | | Rank |
|---|---|---|---|---|---|---|---|---|
| | | [birth_Basic] | [birth_E] | [birth_S] | [shed_Basic] | [shed_E] | [shed_S] | |
| 6 | Cow-to-calf | 0.0% | 0.0% | 4.0% | 1.0% | 0.0% | 3.0% | 7 |
| 5 | Calf-to-calf | 5.9% | 6.9% | 11.9% | 3.0% | 4.0% | 11.1% | 5 |
| 4 | Adult-to-adult contact during the infectious period | 24.8% | 49.5% | 42.6% | 33.3% | 69.7% | 59.6% | 1 |
| 3 | Longer direct contact | 29.7% | 20.8% | 24.8% | 19.2% | 7.1% | 9.1% | 2 |
| 2 | Limited direct contact | 14.9% | 9.9% | 12.9% | 17.2% | 9.1% | 11.1% | 3 |
| 1 | Indirect contact | 16.8% | 7.9% | 3.0% | 19.2% | 7.1% | 5.1% | 4 |
| 0 | No contact | 7.9% | 5.0% | 1.0% | 7.1% | 3.0% | 1.0% | 6 |
| n | Number of isolates for which ancestries could be inferred[a] | 101 | 101 | 101 | 99 | 99 | 99 | |

[a] For 128–101 = 27 isolates (in [birth] scenarios) and for 128–99 = 31 isolates (in [shed] scenarios), no ancestor could be found within the genomic distance threshold of 6 SNPs and, thus, no transmission route could be inferred. Each of these 27 and 31 isolates was the root of a separate transmission tree.

distances of the four dam-daughter pairs in the sample were 0, 25, 26 and 97 SNPs. For the only dam-daughter pair with isolates of identical genotype, only in [birth_S] was the dam assigned as the direct ancestor to her daughter. In all other scenarios dam and daughter were in the same tree, but either separated by 1 or 5 generations, or dam and daughter were in the same generation and assigned to a common ancestor. Inferred ancestries with no epidemiological link indicate the presence of non-sampled or incompletely sampled cows.

## Role of individuals in MAP spread

Seventy percent of cows had only one source of infection; 22%, 7% and 1% had two, three and four different sources, respectively. In all scenarios, transmission of 2–3 genotypes between pairs of cows was recorded, indicative of mixed infections. For nine (39%) of a total of 23 cows with multiple isolates, the algorithm indicated within-host evolution: three cows had 3, and one cow even 4 directly linked isolates. Ancestries within the collection of isolates sampled from the same cow were more frequent in [E] and [S] than in [Basic] scenarios. In [birth] scenarios, each cow infected on average 1.2–1.3 cows, and 46% of animals spread infection to at least one cow, with a maximum of 9–17 recipients. In [shed] scenarios, mean $R_A$ (0.9–1.0), percentage of spreaders (33–42%) and maximum number of recipients (5–7) were lower.

With respect to alternative infection chains due to lack of genomic resolution, the potential number of recipients per cow was on average 1.5 [birth] / 1.3–1.4 [shed], and super-spreaders had up to 10–20 [birth] / 7–10 [shed] potential recipients, depending on the scenario.

Regarding the environment, the same genotype was only twice isolated from the environment and from a cow on the same day. This cow may thus have been the most likely contaminating source. The source of the remaining 20 samples from the farm environment remained undetected. For 11 (50%) isolates from the environment at least one scenario resulted in spill back (or spread by the cow that originally excreted that particular isolate) to 1–6 cows. One environmental sample was even a super-spreader in [shed_E] and [shed_S] scenarios.

## Correlation between reconstructed number of recipients and disease phenotypes

For all scenarios, some correlation between number of recipients per cow and a cow's MAP shedding level, age at first shedding and pattern of infection progress was observed (Table 2). Figures with numbers of recipients for all investigated phenotypes are published in S2 Fig.

**Table 2. Association between number of recipients of an individual cow and her disease phenotype, by scenario.** For phenotypes "shedding level", "age at first shedding" and "infection progress", Spearman's rank correlation coefficients and (*p* values) are presented. For "serostatus", mean number of recipients for both phenotype levels and (*p* values) of Welch's two sample *t* test are presented (n = 57 cows).

| Disease phenotype | Scenarios | | | | | |
|---|---|---|---|---|---|---|
| | [birth_Basic] | [birth_E] | [birth_S] | [shed_Basic] | [shed_E] | [shed_S] |
| **Shedding level[a]** | 0.28 (0.04*) | 0.32 (0.02*) | 0.28 (0.04*) | 0.25 (0.06) | 0.27 (0.04*) | 0.24 (0.08) |
| **Age at first shedding[b]** | -0.20 (0.14) | -0.33 (0.01*) | -0.29 (0.03*) | -0.30 (0.02*) | -0.29 (0.03*) | -0.28 (0.04*) |
| **Infection progress[c]** | 0.32 (0.06) | 0.38 (0.02*) | 0.28 (0.09) | 0.33 (0.06) | 0.34 (0.05*) | 0.26 (0.13) |
| **Serostatus[d]** | 1.0 / 1.7 (0.53) | 1.0 / 2.3 (0.25) | 1.1 / 2.1 (0.24) | 1.1 / 0.5 (0.07) | 0.9 / 0.9 (0.95) | 1.0 / 1.0 (0.93) |

*Good evidence against the null hypothesis of Spearman's rank correlation coefficient = 0 at the significance level of $p < 0.05$.

[a] Shedding level, levels: 0 –always faecal culture negative, 1—low, 2 –high.

[b] Age at first shedding, levels: 0 - ≤3 years, 1 - >3 years, 2—*ante mortem* negative.

[c] Infection progress, levels: 0—*ante mortem* negative, 1—non-progressor, 2 –progressor.

[d] Serostatus, levels: 0—ELISA-negative, 1—ELISA-positive.

There was good to weak evidence ($p$ of 0.02–0.08) that high shedding animals produced more new infections compared to low shedders or cows that were *ante mortem* never tested faecal culture positive. Age at onset of shedding was negatively correlated with number of recipients produced during lifetime: the earlier a cow started to shed, the greater risk for spreading she posed. This correlation was significant ($p$ = 0.01–0.04) for all scenarios except [birth_Basic] ($p$ = 0.14). Regarding infection progress, in all scenarios progressors had the highest mean number of recipients. However, only [E] scenarios showed good evidence for a correlation between pattern of infection progress and number of recipients ($p$ of 0.02 and 0.05). Serostatus was the only investigated phenotype that did not appear to be associated with a risk for spreading. Cows with antibody response, had at least in [birth] scenarios higher numbers of recipients produced compared to animals with only ELISA-negative serum samples. However, no scenario resulted in strong evidence for a more important role of cows with measurable immune response for MAP spread ($p$ of 0.07–0.95). To summarize across all six scenarios, $p$ values were generally smallest for [birth_E] and [shed_E] scenarios, which resulted in significant correlations for three of four investigated phenotypes.

## Discussion

### Key findings

This work presents a method to reconstruct who-infected-whom in an endemic Johne's disease setting, by joining WGS data with explicit longitudinal data. Up to 5 different MAP genotypes could be isolated from individual cows. Genomic distances between isolates were far beyond that expected within-herd and within-host with evolution over time, providing a strong indication for multiple introductions of MAP strains into herds and mixed genotype infections between cows. Reconstruction of transmission trees led to consistent results: cows acquired infection from 1–4 different sources and spread infection to 0–17 recipients, suggesting repeated exposure to shedding animals at different points in time or mixed shedding of one source which led to infection with a heterogeneous inoculum. In the light of low test sensitivities and undetected MAP cases, these numbers should be considered as conservative estimates. For each isolate at least one more likely ancestor could be inferred. For 49% of isolates, even 4 out of 6 infection scenarios resulted in the same choice of ancestor, adding support to the accuracy of reconstructed transmission chains.

Based on comparison of transmission trees and relevant epidemiological constraints, the model that resulted in the best 'who infects whom' answer was the [birth_E] scenario, which included time ordering of isolates based on birthdate of the host and a weighting matrix with number of days that a given susceptible cow spent in the same pen with another cow during her infectious period (Fig 5B). The [birth_E] scenario had not only the highest statistical support for the reconstructed transmission tree among all scenarios with incorporated epidemiology, it also showed the strongest evidence for associations between disease phenotypes of cows and their number of infected recipients. On this farm with a well implemented MAP control program this transmission model favouring horizontal transmission was preferred over other possible models of MAP transmission where weighting preference was given to increased susceptibility in young animals.

Concurrent circulation of dominant strains could be recorded over several years, indicating that some strains were more successful in terms of transmission and infection progression [7, 8]. Other important features of transmission trees were some minor strains that could only be recorded over 2–4 generations of transmissions, and singleton isolates not related to any other isolate. Transmission studies assuming spread of a monophyletic strain will certainly underestimate the complexity of multiple infection chains occurring in parallel.

A particular challenge was inconclusive ancestries for clusters of isolates with identical genotype within the dominant strains. In situations with simultaneous exposure of a susceptible individual to multiple shedders of the same genotype, neither genomic distance nor contact data can be used to resolve ties in ancestries (as in Fig 3). However, the critical question is: is it relevant to know whether cow $X$ or cow $Y$ infected cow $Z$, given that large parts of the operation were perpetually contaminated with one strain? Management-wise, a holistic control strategy would be required, as removing single known shedders could result in limited success in interrupting the infection cycle.

## Mixed infections

Mixed infections are common. This finding adds complexity to the estimation of standard measures in epidemiology, such as the effective reproduction ratio $R$. Despite the decreasing prevalence during the study period, several scenarios resulted in an animal-specific $R_A > 1$. An explanation for this contradiction is that $R_A$ does not consider that some cows acquire infection more than once whereas the proportion of cows that apparently remained MAP negative gradually increased over time. Dependent on this clustering of co-infections with different MAP genotypes, $R_A$ will be considerably larger than the mean number of newly infected recipients generated in the herd ($R_H$) from one generation to the next in the infection chain. This leads to two conclusions: First, infected cows remain susceptible to co-infections with other genotypes. This finding will be relevant for mathematical models: transition between infection states might not be as clear-cut as often assumed. Second, $R_A > R_H$ indicates that some cows have a lower risk of becoming infected than others. How much of these differences can be explained by varying exposure intensity, susceptibility or an alternative explanation was beyond the scope of this study and remains to be investigated.

## Association between disease phenotypes and infection spread

For cows with high levels of shedding, early age at shedding onset and progressive infection, the algorithm returned higher numbers of recipients. These phenotypes measure two concepts: high shedding and exposure over longer time periods as risk factors for MAP transmission. This finding is in line with the literature: super-shedders represent a greater risk for spreading, and close contact with a case and repeated uptake of high doses of bacilli may lead to adult-to-adult infection [4, 5, 11]. Statistical support for this correlation was consistently strong for [E] scenarios weighted by exposure time. This could indicate that duration of exposure to a shedder was a decisive factor for MAP transmission on the investigated farm. Of note, serostatus was the only investigated disease phenotype that appeared not to be associated with risk for spreading. An explanation of this result could be the limited antibody response measured in this herd: only 16% of cows with sequenced MAP isolates were also ELISA-test positive. False-negative test results could have potentially led to misclassification in relation to a cow's true immune response to MAP infection.

The results linked to disease phenotypes presented here should be considered as exploratory; they aimed to show the possibilities of this novel approach to study MAP transmission. A more detailed follow-up is needed to take account of the effects of a changing environment over time, such as reduced infection pressure due to adaptations of the farm management and the decreasing prevalence. Relevant questions in this respect are: what happens if super-shedders are removed: can we see—or even quantify—an effect in the transmission dynamic of the remaining genotypes in response to the control intervention? Will another cow take over the role of a super-spreader or does removing high shedding animals indeed reduce $R$?

## Transmission routes

Adult-to-adult contact during the infectious period of the source cow was the most important transmission route to explain the reconstructed networks. This finding is consistent with Schukken et al. [8], who showed that cows that were infected with a particular MAP strain were significantly more exposed as adults to cows shedding the same strain compared to cows that were culture negative for MAP at slaughter. Cow-to-calf and calf-to-calf contacts during early life accounted for less than 1 in 12 transmissions. These analyses thus did not support the hypothesis that dam-daughter infections were the principal transmission route on the farm under investigation. This farm had implemented rigid interventions for MAP control, daughters were separated from dams as quickly as possible and contact between age groups was limited. This could explain the minor role of dam-newborn calf and calf-to-calf transmissions and indicate success of the implemented interventions. However, MAP exposure during adulthood appeared to be sufficient to maintain infection over years. To confirm this hypothesis, the analysis needs to be repeated for farms with different calf management protocols to investigate correlation between interventions and contribution of transmission routes for long-term persistence of MAP.

## Strengths of this study

In the presence of within-host diversity, analysis of a single isolate will miss clonal, closely related or distant strains shared between the source and its recipients, resulting in inaccurate conclusions about transmission. Analysis of several isolates per cow, and also per sample, is important for source tracking and status determination of the animals involved [20]. Particularly with pathogens with slow divergence, uncertainty will remain even if all genotypes are observed, as individual transmission routes cannot be resolved by sequence data alone [23]. The conclusions of this study were based on a unique dataset that allowed the study of individual transmission networks of genotypes and diversification of the MAP population within-host and within-herd over time and in many facets. Only the combination of precise longitudinal data on infection status, detailed demographic data and bacterial strain type information of a densely sampled population made it possible to trace infection in such detail.

## Choice of method

SeqTrack has limitations regarding inference of the underlying transmission process and timing of exposure. However, the algorithm supports incorporation of all kinds of data to describe proximity in the form of weighting matrices, a criterion we prioritized in the assessment of available methods to apply in our study as strain diversification was expected to be limited. SeqTrack proved to be sufficiently versatile to be expanded for investigations of endemic disease. On purpose, no assumptions about transmission routes were made, which enabled independent evaluation of reconstructed networks for transmission routes. Weighting by exposure time [E] was based solely on explicit data and weighing by susceptibility [S] was based on accepted knowledge of MAP epidemiology. Of note, scenarios informed by data and knowledge resulted in more consistent ancestries compared to scenarios based solely on genomic data. To allow for ancestries with short time-intervals between two transmission events, no minimum duration of pre-infectious period was entered. Hence, transmission routes for scenarios in which time between MAP uptake and shedding is assumed to be short, such as calf-to-calf transmission or pass-through shedding of adults, could be investigated.

   An apparent limitation of SeqTrack is its strong dependence on temporal ordering of isolates. The algorithm does not clearly account for the potentially long delay between unobserved exposure and observed sampling events. Due to the long incubation period of MAP, the order of ancestry cannot be inferred with certainty. For example, mutated genotypes might

have been sampled before their ancestor. Other approaches such as Bayesian algorithms will be superior in the inference of the underlying transmission process and timing of exposure. Nevertheless, limiting the problem to more exact inference of time of exposure of recipients would clearly underestimate the complexity of MAP epidemiology. Even if we knew the time point of infection, we would still face the challenge of the long incubation period, lesion formation and open questions on the role of pathogen-host immunity interplay for the start of bacterial shedding of the recipient cow [35]. Therefore, we chose an approach where we investigated with three birth and three shed scenarios, two opposing temporal indicators as proxies for the time of infection and time of becoming infectious. It could be shown that the order of isolates within an infection chain was time dependent. However, isolates were assigned to the same infection chain, independent of their date variable.

## Limitations

For this study, although it was based one of the most detailed MAP cohorts published to date, ideally (at least) one more generation of cows should have been sampled to reduce the impact of both right- and left-censored data on the results. As MAP prevalence decreases, sampling effort increases even more compared to each WGS sequence gained in return. In this study, on average 33 faecal, tissue or environmental samples were cultured per sequenced isolate; towards the end of the study this number increased towards 100 samples for one recovered sequence. Particularly for studying low prevalent MAP, field studies will quickly reach their resource limit (in the absence of diagnostic methods to identify MAP infected individuals more efficiently). However, only field data can show the true variation in incidence, change of prevalence and effects of control interventions on transmission in a non-controlled environment.

Although all cows were sampled periodically, low test sensitivity will have led to unobserved infected cows or additional MAP genotypes shed by known MAP-shedders. Since infection source is only attributed to sampled isolates, adding such non-detected isolates to the analysis could either result in new common sources of infection for separate transmission trees or cause additional generations between (wrongly) inferred direct ancestries within a transmission tree. Nevertheless, the original isolates would remain in the same transmission tree. The reproductive ratios at genotype level $R_{GT}$ would have been higher; the above presented values thus represent estimates at the low end. With more cows potentially contributing to MAP spread, the $R_A$ would have varied for individual cows. Assuming that all genotypes were affected to a similar extent by the non-perfect test characteristics, the authors believe that the main conclusions of this work remain valid.

As youngstock were not sampled, no genomic data to directly confirm calf-to-calf or adult-to-calf transmission were available. From the data it could not be determined whether genotypes that led to the initial infection were among the available collection of isolates from a cow.

SeqTrack is highly sensitive to genomic distance between isolates. SNP distances may vary for technical reasons, and any errors leading to different calls at SNP positions could potentially lead to different ancestries. Assuming similar error rates across isolates, the relevant contributors to infection spread could nevertheless be construed. When comparing maximum parsimony across transmission trees, it should be kept in mind that the maximum genomic parsimony is likely present in the model without any epidemiological limitations. As a consequence, the decision on the best fitting model is based on a combination of epidemiological and biological knowledge and maximum parsimony within this defined epidemiological and biological transmission framework. Actually, a similar parsimony in models with epidemiological and biological constraints and the basic model without these constraints is an outcome that heavily favours the outcome of the model with the constraints.

A distance threshold of 6 SNPs was used as a cut-off to rule in/out direct ancestry between isolates. This approach was applied to several pathogens, including tuberculosis in humans for which the most commonly employed cut-off is based on the finding that epidemiologically linked patients were genomically linked by ≤5 SNPs, with an upper bound of 12 SNPs between any two linked isolates [36]. Thresholds for discriminating mixed infections from within-host evolution still lack standardization [37]. Even when this analysis was largely insensitive to the mutation rate, it needs to be kept in mind that rates of evolution (and as a result also distance thresholds) may differ across lineages of the same species [22].

## Conclusions

Mixed infections of dairy cows with MAP appear to be common, and some strains are more successful than others in terms of transmission. The within-host MAP diversity observed in this study is among the highest ever reported. To the best of the authors' knowledge, instances of more than 5 genotypes sampled from a cow have not been described in the literature thus far. Cows infected with one or more strains remain susceptible to infections with other MAP genotypes. Transmission studies are therefore expected to benefit from strain-specific transmission parameterisation. To be able to observe the full range of diversity in samples with heterogeneous MAP populations, methods for pathogen isolation are needed which support detection and quantification of multiple genotypes. This work presents a method for reconstructing "who infects whom" based on genomic data with greater epidemiological and statistical support. Reconstructed infection chains confirmed high shedding and exposure to shedders over longer time periods as risk factors for MAP transmission on the investigated dairy farm. We believe that the method will be useful for further studies on the relevance of transmission routes and role of individuals expressing distinct disease phenotypes in infection dynamics of endemic disease.

WGS is invaluable in studying pathogen transmission, both with outbreaks and in endemic settings. However, WGS is not a solution for low test sensitivity which leads to non-observed isolates. In addition, especially for MAP-like pathogens the question remains, when does an animal become infected and infectious? The method presented in this work is able to indicate where infection cycles went undetected. This information can be used to adapt sampling to better capture underlying transmission processes. Knowledge of pathogen biology and availability of precise longitudinal data are crucial to maximise benefits of WGS and validly reconstruct infection chains.

## Supporting information

**S1 Fig. Reconstructed transmission tree of three scenarios (n = 128 isolates).** (A) [shed_Basic], (B) [shed_E], (C) [shed_S]. Isolates sampled from the same cow are labelled with successive numbers and are shown in vertices of same colour and outline. White vertices represent cows with only one isolate. Dark green vertices (labelled 99–120) represent environmental samples. Edge labels and edge colour indicate number of SNPs difference between ancestor and descendant.
(PDF)

**S2 Fig. Numbers of recipients produced by individual cows.** Boxplots with numbers of recipients produced by individual cows, by disease phenotype and scenario. (A) shedding level (0 – always faecal culture negative, 1—low, 2 –high), (B) age at first shedding (0 - ≤3 years, 1 - >3 years, 2—ante mortem negative), (C) infection progress (0—ante mortem negative, 1—non-progressor, 2—progressor), (D) serostatus (0—ELISA-negative, 1—ELISA-positive). Scenarios

(from left-most to right-most column): [birth_Basic], [birth_E], [birth_S], [shed_Basic], [shed_E] and [shed_S].
(PDF)

**S1 Table. Overview of unique identifiers of MAP isolates and cows.** Column A lists the isolate number used in Fig 5. Column B lists the isolate label used in the S1 File in fasta-format. Column C lists the "sample name" provided to NCBI's Sequence Read Archive database, where all isolates used in this article are publicly deposited under the following two BioProject accession numbers: PRJNA725521 and PRJNA686527. Column D lists the eartags of the cows from which the MAP isolate was sampled. Note: 1–5 MAP isolates could be sampled from individual cows.
(XLSX)

**S2 Table. Statistical support for inferred ancestries, by scenario.** Columns H—M show the genomic distance between ancestor and descendant, measured in number of SNPs between each pair of isolates. Columns N—S show the statistical support for each inferred ancestry, expressed as a *p*-value and calculated based on maximum likelihood. Summary statistics are presented at the bottom of the table. Descendants with no ancestors can be identified as singleton isolates in the figures of the reconstructed transmission trees Fig 5 and S1 Fig. Singleton isolates have >6 SNP difference to any other isolate).
(XLSX)

**S1 File. SNP alignment of 150 MAP sequences with 1,472 SNPs of the core genome.**
(FASTA)

**S2 File. R code of analysis.**
(R)

## Acknowledgments

The authors thank all staff and veterinarians of the participating farm for their invaluable input and support in collecting samples and data. We also acknowledge the teams of collaborators who performed all samplings on the farm and performed MAP cultures in the course of this large cohort study. We also would like to acknowledge Mart de Jong for his valuable comments on analyses.

## Author Contributions

**Conceptualization:** Annette Nigsch, Ynte H. Schukken.

**Formal analysis:** Annette Nigsch, Tod P. Stuber.

**Funding acquisition:** Yrjö T. Gröhn.

**Investigation:** Paulina D. Pavinski Bitar.

**Methodology:** Annette Nigsch.

**Resources:** Suelee Robbe-Austerman.

**Software:** Tod P. Stuber.

**Supervision:** Ynte H. Schukken.

**Writing – original draft:** Annette Nigsch, Ynte H. Schukken.

**Writing – review & editing:** Yrjö T. Gröhn, Ynte H. Schukken.

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
