## [Decision Letter · Decision Letter 0]

15 Mar 2021

PONE-D-21-03120

Who infects Whom? - Reconstructing infection chains of Mycobacterium avium ssp. paratuberculosis in an endemically infected dairy herd by use of genomic data

PLOS ONE

Dear Dr. Nigsch,

Thank you for submitting your manuscript to PLOS ONE. After careful consideration, we feel that it has merit but does not fully meet PLOS ONE’s publication criteria as it currently stands. Therefore, we invite you to submit a revised version of the manuscript that addresses the points raised during the review process.

This is a very interesting manuscript. Both reviewers are very supportive and I really enjoyed reading it. However, the reviewers have raised some very interesting points that would improve this manuscript further. Please revise the manuscript accordingly.

We look forward to receiving your revised manuscript.

Kind regards,

Angel Abuelo, DVM, MRes, MSc, PhD, DABVP (Dairy), DECBHM

Academic Editor

PLOS ONE

Journal Requirements:

2. We note that you are reporting an analysis of a microarray, next-generation sequencing, or deep sequencing data set. PLOS requires that authors comply with field-specific standards for preparation, recording, and deposition of data in repositories appropriate to their field. Please upload these data to a stable, public repository (such as ArrayExpress, Gene Expression Omnibus (GEO), DNA Data Bank of Japan (DDBJ), NCBI GenBank, NCBI Sequence Read Archive, or EMBL Nucleotide Sequence Database (ENA)). In your revised cover letter, please provide the relevant accession numbers that may be used to access these data. For a full list of recommended repositories, see http://journals.plos.org/plosone/s/data-availability#loc-omics or http://journals.plos.org/plosone/s/data-availability#loc-sequencing.

Reviewers' comments:

Reviewer's Responses to Questions

**Comments to the Author**

1. Is the manuscript technically sound, and do the data support the conclusions?

Reviewer #1: Partly

Reviewer #2: Yes

2. Has the statistical analysis been performed appropriately and rigorously? 

Reviewer #1: I Don't Know

Reviewer #2: Yes

3. Have the authors made all data underlying the findings in their manuscript fully available?

Reviewer #1: Yes

Reviewer #2: No

4. Is the manuscript presented in an intelligible fashion and written in standard English?

Reviewer #1: Yes

Reviewer #2: Yes

5. Review Comments to the Author

Reviewer #1: The manuscript submitted by Nigsch et al. for review in PLOS-ONE describes the use of a strategy employing genome sequencing, SeqTrack and network analysis to construct lineage and transmission trees Mycobacterium avium ssp. paratuberculosis (MAP) isolated form a diary heard from New York. Some questions/comments raised during the current review are as follows:

(1) It was good to see that the lineage/transmission analysis using different scenarios (assumptions) led to trees with similar isolates and outcomes, but some points regarding data and sample collection were not clear. According to the methods section, the MAP isolates sequenced were collected between 2004-2008 (line 168). In lines 126-129 it is mentioned that this was a retrospective study based on available sequenced MAP isolates. Was the sequencing conducted as part of the current study or a previous (separate) one and were the isolates bio-banked, etc.? Also, are the sequences publicly available?

(2) Some of the figures and legends might need clarification. For example, Fig 3 is only referenced once (line 325) in the whole manuscript to show two alternatives leading to isolates 116 and 114, respectively. Also, there are no different edge colours in the specific figure. Similarly, it is hard to see details in figure 5. Therefore, I recommend revisiting figures and making them and the legends more descriptive if required.

(3) Line 707-708: The authors mention that “5 MAP genotypes form a sampled cow has not been reported previously”. Although this might be correct, Podder et al (2015, PLoS One. 10(4): e0126071) did report on MAP mixed strain infections (although not using genomics) and showed that when they analyzed up to 5 isolates, they could see variations. They also highlighted the need to revisit the numbers of isoaltes analyzed for source tracking and epidemiological studies, might be a good idea to reference or comment?

(4) The discussion is long and seems to go beyond the scope of the data presented at times. I would recommend pairing it down to the main findings. Also, some of the points raised on mixed strain infections and intra host evolution were discussed in a recent systematic review by Byrne et al. (2020, Frontiers in Genetics. 11:600692). Might be a good idea to reference or comment?

Reviewer #2: An extremely well written and carefully argued manuscript addressing an issue of critical importance to this particular disease. The application of these methods that will be increasingly important within this discipline of infectious disease epidemiology in animal health.

This methodology, not previously used in the field of JD, offers some extremely valuable insights into the epidemiology of JD in this setting. Key findings include:

- at a herd level, multiple strains and multiple introductions

- at an individual level, evidence of mixed infections and ongoing susceptibility to co-infection

- with respect to transmission, further evidence of the role of superseders, further evidence in support of adult-to-adult infection, and in this setting very limited cow-to-calf transmission.

This work, and indeed the broader work from this group, raises important questions with respect to current understanding of the epidemiology of Map in cattle. In particular, the findings of predominant adult-to-adult infection and the very limited contribution of cow-to-calf transmission. It will be very helpful, in future research, to clarify the relative importance of different transmission routes under different management situations. Here, the authors suggest that the limited contribution of cow-to-calf transmission is a consequence of very high levels of herd management, which certainly seems plausible.

The Discussion is excellent, with the authors reflecting on the study findings, the study strengths/limitations and the methodology with considerable care and insight. I support the authors suggestion of this work as exploratory and needing more detailed follow-up, and would strongly support this. Further, the authors propose methodology to do this, and some key questions to be resolved

I have no substantive comments, other than to congratulate the authors for this excellent work. And I greatly look forward to further work from this group on this important issue.

Several minor editorial issues:

- 123 and 124, note (7) and (8) have been repeated

- 283, a missing closing bracket

6. PLOS authors have the option to publish the peer review history of their article (what does this mean?). If published, this will include your full peer review and any attached files.

Reviewer #1: No

Reviewer #2: No

---

## [Author Response · Author response to Decision Letter 0]

29 Apr 2021

Rebuttal 

Our responses to the comments of referee 1 and referee 2 are given in red font in the document "response to the reviewers".

Note on line numbers: when we accepted all track changes to produce the “Manuscript” from the “Revised manuscript with track changes”, we noted that line numbers changed. The manuscript has thus 839 lines. The “Revised manuscript with track changes” shows 942 as last line number, instead of the correct number of 839 lines. In our rebuttal below, we always refer to the line numbers in the version with track changes. 

We apologize for this inconvenience, but this seems to be an error of Word, which we could not solve (we tried several approaches). 

Academic editor

We thank the academic editor for supporting our work and for the invitation to submit a revised version.

Reviewer #1

The manuscript submitted by Nigsch et al. for review in PLOS-ONE describes the use of a strategy employing genome sequencing, SeqTrack and network analysis to construct lineage and transmission trees Mycobacterium avium ssp. paratuberculosis (MAP) isolated form a diary heard from New York. Some questions/comments raised during the current review are as follows:

(1) It was good to see that the lineage/transmission analysis using different scenarios (assumptions) led to trees with similar isolates and outcomes, but some points regarding data and sample collection were not clear. According to the methods section, the MAP isolates sequenced were collected between 2004-2008 (line 168). In lines 126-129 it is mentioned that this was a retrospective study based on available sequenced MAP isolates. Was the sequencing conducted as part of the current study or a previous (separate) one and were the isolates bio-banked, etc.? Also, are the sequences publicly available?

Response: The sequencing was part of the current study (information added in L162). All isolates were bio-banked and are available on request. The sequences were deposited on NCBI and are publicly available (information added in L169-173).

(2) Some of the figures and legends might need clarification. For example, Fig 3 is only referenced once (line 325) in the whole manuscript to show two alternatives leading to isolates 116 and 114, respectively. Also, there are no different edge colours in the specific figure. Similarly, it is hard to see details in figure 5. Therefore, I recommend revisiting figures and making them and the legends more descriptive if required.

Response: We added a second reference to Fig 3 in the discussion (L539) to show the impact of the issue of alternative infection chains on disease control. We improved the colour contrast of edge colours in Fig 3. 

With Fig 5, we refrained from adding more additional graphical elements, but still wanted to keep all information in the figure. As a solution to provide the reader with more details in an improved format, we added information to the S2 Table in the supporting information. This S2 Table was mentioned in the legend of Fig 5 (L419-420).

(3) Line 707-708: The authors mention that “5 MAP genotypes form a sampled cow has not been reported previously”. Although this might be correct, Podder et al (2015, PLoS One. 10(4): e0126071) did report on MAP mixed strain infections (although not using genomics) and showed that when they analyzed up to 5 isolates, they could see variations. They also highlighted the need to revisit the numbers of isoaltes analyzed for source tracking and epidemiological studies, might be a good idea to reference or comment?

Response: We thank reviewer 1 for highlighting the results of the article of Podder et al., 2015. We included their findings in our introduction (L77) and in our discussion (L597-598).

(4) The discussion is long and seems to go beyond the scope of the data presented at times. I would recommend pairing it down to the main findings. Also, some of the points raised on mixed strain infections and intra host evolution were discussed in a recent systematic review by Byrne et al. (2020, Frontiers in Genetics. 11:600692). Might be a good idea to reference or comment?

Response: We shortened the discussion from originally 205 lines to 169 lines, which equals a reduction of ca. 18%. Thereby we attempted to keep those parts that discussed the main epidemiological findings. Due to the cuts, we had to make minor adaptions and some shifts in the text to keep the text flow. All these adaptions can be retraced in the text copy with track changes.

We thank reviewer 1 for recommending the recently published article by Byrne et al., 2020. We referenced this work in L759. 

Reviewer #2: 

An extremely well written and carefully argued manuscript addressing an issue of critical importance to this particular disease. The application of these methods that will be increasingly important within this discipline of infectious disease epidemiology in animal health.

This methodology, not previously used in the field of JD, offers some extremely valuable insights into the epidemiology of JD in this setting. Key findings include:

- at a herd level, multiple strains and multiple introductions

- at an individual level, evidence of mixed infections and ongoing susceptibility to co-infection

- with respect to transmission, further evidence of the role of superseders, further evidence in support of adult-to-adult infection, and in this setting very limited cow-to-calf transmission.

This work, and indeed the broader work from this group, raises important questions with respect to current understanding of the epidemiology of Map in cattle. In particular, the findings of predominant adult-to-adult infection and the very limited contribution of cow-to-calf transmission. It will be very helpful, in future research, to clarify the relative importance of different transmission routes under different management situations. Here, the authors suggest that the limited contribution of cow-to-calf transmission is a consequence of very high levels of herd management, which certainly seems plausible.

The Discussion is excellent, with the authors reflecting on the study findings, the study strengths/limitations and the methodology with considerable care and insight. I support the authors suggestion of this work as exploratory and needing more detailed follow-up, and would strongly support this. Further, the authors propose methodology to do this, and some key questions to be resolved

I have no substantive comments, other than to congratulate the authors for this excellent work. And I greatly look forward to further work from this group on this important issue.

Several minor editorial issues:

- 123 and 124, note (7) and (8) have been repeated

- 283, a missing closing bracket

Response: We thank reviewer #2 for the very positive feedback and support of the ideas of our manuscript. We hope that our colleagues out there will also enjoy reading the published version of this manuscript. 

The editorial issues were corrected. 

Deposition of data in repositories 

Sequence read data have been deposited at NCBI in the Sequence Read Archive database under the following two BioProject accession numbers: PRJNA725521 and PRJNA686527. For completeness, the final snp data set used in this work has been added as S1 file. A table was added to the Supporting information with unique identifiers to match the SNP alignments of S1 file with the isolates deposited at NCBI.

Changes to the Supporting information

S1 Table was added. This table gives an overview on unique identifiers of isolates, SNP alignments and cows. With this table, isolates deposited at NCBI, SNP alignments and cows can be matched. 

The original S1 Table is now titled S2 Table and contains extra information on the pairwise genomic distance between ancestors and descendants. This information supports the reader to note the details in Fig 5 and S2 Fig.

Changes to the reference list

Based on comments of reviewer 1, we cited two additional articles: 

20. Podder MP, Banfield SE, Keefe GP, Whitney HG, Tahlan K. Typing of Mycobacterium avium subspecies paratuberculosis isolates from Newfoundland using fragment analysis. PloS one. Public Library of Science; 2015;10(4):e0126071.

37. Byrne AS, Goudreau A, Bissonnette N, Shamputa IC, Tahlan K. Methods for detecting mycobacterial mixed strain infections-a systematic review. Frontiers in Genetics. Frontiers; 2020;11:1590.

A third article was still “in press” when we submitted our original version of this manuscript to PLOS ONE. Meanwhile, the article was published. 

30. Richards VP, Nigsch A, Pavinski Bitar P, Sun Q, Stuber T, Ceres K, et al. Evolutionary Genomic and Bacterial Genome-Wide Association Study of Mycobacterium avium subsp. paratuberculosis and Dairy Cattle Johne’s Disease Phenotypes. Applied and Environmental Microbiology. Am Soc Microbiol; 2021;87(8).

---

## [Editor Report · Decision Letter 1]

3 May 2021

Who infects Whom? - Reconstructing infection chains of Mycobacterium avium ssp. paratuberculosis in an endemically infected dairy herd by use of genomic data

PONE-D-21-03120R1

Dear Dr. Nigsch,

We’re pleased to inform you that your manuscript has been judged scientifically suitable for publication and will be formally accepted for publication once it meets all outstanding technical requirements.

Kind regards,

Angel Abuelo, DVM, MRes, MSc, PhD, DABVP (Dairy), DECBHM

Academic Editor

PLOS ONE
---

## [Editor Report · Acceptance letter]

5 May 2021

PONE-D-21-03120R1 

Who infects Whom? - Reconstructing infection chains of *Mycobacterium avium* ssp. *paratuberculosis* in an endemically infected dairy herd by use of genomic data 

Dear Dr. Nigsch:

I'm pleased to inform you that your manuscript has been deemed suitable for publication in PLOS ONE. Congratulations! Your manuscript is now with our production department. 

Kind regards, 

on behalf of

Dr. Angel Abuelo 

Academic Editor

PLOS ONE